# Performance Indicators of Printed Construction Materials: a Durability-Based Approach

**Zoubeir Lafhaj \* and Zakaria Dakhli** 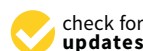

Civil Engineering, Centrale Lille, 59651 Cité Scientifique, France; zakaria.dakhli@centralelille.Fr
\* Correspondence: zoubeir.lafhaj@centralelille.fr; Tel.: +33-3-20-33-54-87

**Abstract:** Studying the durability of materials and structures, including 3D-printed structures, is now a key step in better meeting the challenges of sustainable development and integrating technical and economic aspects from the design phase into the execution phase. While digital and robotics technologies have been well developed for construction 3D printing, the material aspect still faces critical issues to meet the evolving requirements for buildings. This research aims to develop performance indicators for 3D-printed materials used in construction regardless of the nature of the material. A general guideline is to be established as a result of this research. Thus, the literature review analyzes traditional durability approaches to construction materials and challenges are identified for potential applications in construction. The results suggest that performance indicators for 3D-printed materials should be checked as printable through an experimental case study. This research could be of interest to researchers, professionals, and start-ups in the construction and materials research fields.

**Keywords:** 3D printing; recycled materials; construction; building; material; performance indicators; printability

---

## 1. Introduction

Durability is integrated into a sustainable development approach in terms of environmental preservation, technical and economic optimization of structures, and control of construction and maintenance costs [1,2]. Understanding the durability of materials is crucial in construction projects. It guides the design of the structure, the formulation of the material, and its implementation. Sustainability has been defined by Eurocode 2 as follows:

"A durable structure must meet the requirements of serviceability, strength, and stability throughout the life of the project, without significant loss of functionality or excessive unexpected maintenance."

The main objective of analyzing the durability of a material is to precisely select the desired characteristics in order to optimize its composition according to the environmental constraints to which it will be subjected during expected service life.

The use of cementitious materials in construction is vast and diversified [3]. Indeed, the properties and characteristics of cementitious materials have allowed them to evolve in line with scientific discoveries and industrial progress [4]. Nowadays, several researchers are interested in printing cementitious material [5] and several factors come into play and should be taken into account to ensure printability [6]. Materials printability is still one of the major challenges [7]. There is no relevant guideline for the 3DCP in terms of material formulation and no standard criteria are set to limit the specifications [8]. Thus, various efforts must be made to standardize the technology in the construction industry.

In this light, the durability of cementitious materials derived from additive manufacturing technology is studied in this research. The key challenges for printability were outlined by Wangler

et al. [9]. The focus was the concrete extrusion and intermix with the previously deposited layer. Each layer must support its own weight and the weight of the material to be subsequently deposited.

## 2. Normative and Regulatory Contexts

Concrete structures have comprehensive normative support in the form of European and French standards, which makes it possible to better understand and control the durability of structures.

Standard NF EN 206 (Concrete—specifications, performance, production, and conformity) states the requirements for concrete components, its properties in fresh and hardened state, limitations imposed on its composition, specification, delivery of fresh concrete, production control procedures, conformity criteria, and evaluation.

Two different approaches are presented in this standard: a prescriptive approach and a performance-based approach. They specify, in terms of composition and performance, concrete formulas adapted to each exposure class. The obligation of means and the obligation of results are the two alternatives used to study the durability of the material in its environment. The requirements for each approach are different.

### 2.1. Prescriptive Approach

This approach presents the traditional approach to studying the durability of concrete. It defines the specified limit values to meet the requirements of durability in relation to a proposed use under certain environmental conditions.

The NF EN 206 standard defines specified limit values for the composition and properties of concrete according to each exposure class in two tables (NA.F.1 and NA.F.2) [10]. These values are based on 50-year service life of the structure.

Depending on each exposure class, the standard specifies limit values for:

- The maximum water-efficiency/equivalent binder ratio,
- The minimum strength class of the concrete, the minimum content of equivalent binder, and the minimum air content.

It also contains requirements regarding additions and the maximum quantities allowed for the calculation of the Liantéq with each addition (fly ash, silica fume, ground slag, limestone, or siliceous addition), and for each type of cement to be used.

The quantity of equivalent binder (Liantéq) corresponds to the quality of cement (*C*) increased by the quantity of addition (*A*) weighted by a coefficient (*k*) according to each type of addition (Liantéq = *C* + *k*\**A*).

This prescriptive approach has shown its limitations in assessing the potential durability of concretes. Indeed, several studies have shown that compositional limit values do not make it possible to estimate the durability of a structure [4,11].

In addition, this approach limits the use of non-standardized components and the number of additives. This may also represent a brake on the development of new materials for 3D printing technology given the properties required for this type of application.

In addition to this prescriptive approach, a high-performance approach is authorized by the NF EN 206 standard that offers the only alternative for qualifying different or innovative systems. It is necessary to guide or optimize composition choices according to the desired durability by taking into account technical, economic, and environmental aspects as shown by Hooton and Bickley [12]. The durability of concretes is also understood by considering certain characteristics or properties of the material that are known to be of interest in predicting its evolution when exposed to specific environmental conditions [13].

### 2.2. Performance Approach

The performance approach is a powerful and necessary lever for innovation and sustainable development. A performance-based approach is necessary for the development and improved durability of new materials as they can only be qualified based on their compositions and their performance and behavior in a given environment. It also makes it possible to consider the use of local materials, mineral additions, and additives as new degrees of freedom to address technical, environmental, and economic issues. Various studies have examined the impact of alternative constituents on cement and natural aggregates on the performance and durability of concrete. For example, Jimenez et al. and Paine and Dhir studied the effect of recycled aggregates by using a performance-based approach [14,15]. The national RECYBETON project addressed the issue of the use of recycled aggregates from deconstructed concrete [16].

A great deal of interest has been shown in the performance-based approach in the context of the national PERFDUB research project [17]. The objective is to define a methodology at the national level to justify the durability of concretes. The aim is to aggregate knowledge and feedback and to fill gaps in a framework that brings together all the actors concerned so that an effective approach becomes operational and widely used.

Building on previous and ongoing projects (e.g., [18–20]), different concepts have been used to implement a successful approach to sustainability. The two main concepts correspond, on the one hand, to the method based on sustainability indicators and, on the other hand, to the system based on the use of performance tests.

### Durability Indicators

Durability indicators are parameters that appear to be fundamental in assessing and predicting the durability of the material and structure with respect to the degradation process under consideration [21]. These are essential tools to anticipate damage and optimize maintenance.

These indicators make it possible to determine the properties of materials in relation to the environment and to feed predictive models for aging.

The AFGC guide summarizes the different methods available for determining durability indicators as shown in Table 1. Example of research projects on tensile strength of 3D printed materials for construction [22–24]. The concept of printability was also investigated in literature research, and linked to some indicators, such as the shear rate, viscosity, and thixotropy [6,25].

**Table 1.** Methods for measuring durability indicators.

| | Parameters to be Determined | Method | Deadline for the Result | Duration of the Test | Precision | Cost Evaluation | Observation |
|---|---|---|---|---|---|---|---|
| **General durability indicators** | Water porosity | Hydrostatic weighting | 15 days | 3.5 months | 1.5% | * | |
| | Apparent or effective chloride diffusion coefficient | Migration in a steady state | 15 days | 4 months | 15% of the average value | ** | |
| | | Migration in a non-stationary mode | 1 week | 3.5 months | 15% of the average value | ** | |
| | | Diffusion in a non-stationary mode | 3 months | 6 months | 15% of the average value | *** | |
| | Gas permeability | CEMBUREAU | 45 days | 4.5 months | 30% of the average value | ** | Specific equipment |
| | Permeability to liquid water | Pressurized water permeameter (NFP 18-855) | 15 days | 3.5 months | 1 order of magnitude | * | |
| | CaOH$_2$ content | ATG | 1 week | 3.5 months | 1.5% | ** | Specific equipment |
| | | Chemical analysis | 1 week | 3.5 months | 2% | * | |
| **Parameters required for the application of indirect methods** | Characteristics of the porous structure | Mercury intrusion measurements | 15 days | 3.5 months | 1.5% | ** | Specific equipment |
| | Electrical resistivity | [ANDR01] | 1 week | 3.5 months | 10% of the average value | * | |
| | Isotherms of water vapor sorption | Methods of saturated saline solutions (LPC n°58) | 6 months | 9 months | 10% of the average value | *** | |
| | Isotherms of interaction matrix–chlorides | Ex. Immersion | 2 months | 5 months | 10% of the average value | ** | |
| **Alkali reaction–Specific indicators** | Quantity of silica released by aggregates as a function of time | Cinetic test NFP 18-589 or modified cinetic test NFP 18-594 | 1 week | 1 to 2 weeks | 10% of the average value | ** | |
| | Balance of the alkalis in the concrete formula | LPC n°17 and 48 | 1 week | 1 week | 0.1 | ** | |
| | Swelling deformation | Project NFP 18-454 | 5 months | 5 months | ±20 (μm/m) | *** | |

Sustainability indicators are divided into two categories according to the AFGC guide, "Concrete design for a given life span of structures" [26]:

- General durability indicators valid for different types of degradation (corrosion of reinforcements, alkali reaction...).
- Sustainability indicators specific to a given degradation process, such as alkali reaction or freezing.

In addition, the direct determination of some general sustainability indicators may be replaced by the direct determination of alternative indicators.

General durability indicators are key parameters for the durability of concrete with respect to corrosion of reinforcement, alkali reaction, or any other degradation. The general indicators defined are as follows [26]:

- Porosity accessible to water,
- Diffusion coefficient (apparent or effective) of chloride ions,
- Gas permeability,
- Permeability to liquid water, portlandite content $Ca(OH)_2$.

The determination of all these parameters is not systematically necessary as it depends on the foreseeable damage to the environment and the practical case studied. The values of the durability indicators vary greatly with the age of the material before three months, especially when the concrete formula contains a high proportion of hydraulic or slow-reaction pozzolanic mineral additions (fly ash, slag). Another important parameter in determining sustainability indicators is the water status of the samples, which is essential for the development of chemical reactions and their macroscopic consequences [27–29]. The saturation rate or water vapor sorption and desorption isotherms present the complementary parameters necessary to determine and interpret various sustainability indicators [30].

Several studies have shown the impact of water status on concrete properties. For example, the compressive strength of dry concrete increases by 40–70% compared with saturated concrete [31]. On the other hand, various studies have shown the influence of water status on transfer parameters:

- Gas permeability increases when the average water saturation rate decreases [17,18].
- Permeability to liquid water varies according to the saturation rate. Figure 4 shows the evolution of water permeability as a function of saturation rates. It shows that for S ≤ 40%, the transport in the liquid phase is negligible, whereas, for the S > 80% domain the increase in relative permeability is very significant [32].
- For the penetration of chloride ions, the diffusion coefficient decreases with the water content [33,34].

It is interesting to determine general sustainability indicators under conditions that approximate *in situ* water conditions of a structure (RH > 60%).

## 3. 3D Printing for Construction

3D printing in construction is a new field of study that has attracted many researchers [3,35,36]. The current challenges of 3D printing concern the robotics systems, the software part [9], and the material part [3,37]. The principle of 3D printing in construction has many similarities with the one we already know for ordinary printing, a nozzle deposits layers of viscous concrete at each pass by climbing each time a notch [22]. One of the major scientific locks of construction 3D Printing is the material. The latter passes through the mixer, pump, robot and then is extruded through the nozzle. The printed material should be designed to withstand horizontal and vertical constraints throughout the printing process [38]. The main material currently used for 3D printing is cement. One of the main advantages of printing a cementitious material is avoiding the laborious formwork phase, which represents 35–60% of the total cost of concrete structures.

The material is initially in a viscous state and solidifies once printed. The 3D-printed material stands in place and reaches a strength that allows it to resist the weight of the layers that will come on top.

## 4. Research Vision

The objective of this research is to define a methodology for assessing the durability of 3D-printed materials.

The performance-based approach is an appropriate method because the materials developed are of a specific composition that should meet the specified requirements for fresh and cured concrete while considering the production process and the 3D printing process (Figure 1) chosen for executing the work. The prescriptive approach, however, does not meet all the criteria for such a goal.

In addition, the performance indicators for the 3D printing process in construction aim to promote sustainable concrete formulas with low environmental impacts and to increase the use of additives and admixtures to improve certain properties, whether fresh, during setting and curing, or in a cured state. Indeed, these additives mainly impact the rheology, hydration kinetics, and mechanical performance [22,39].

Studying the durability of materials and structures, including 3D-printed structures, is now a key step in better meeting the challenges of sustainable development and in integrating technical and economic aspects from the design phase into the execution phase. This work makes it possible to provide answers and recommendations to prevent possible damage to the structures and to estimate their lifespan using existing predictive models.

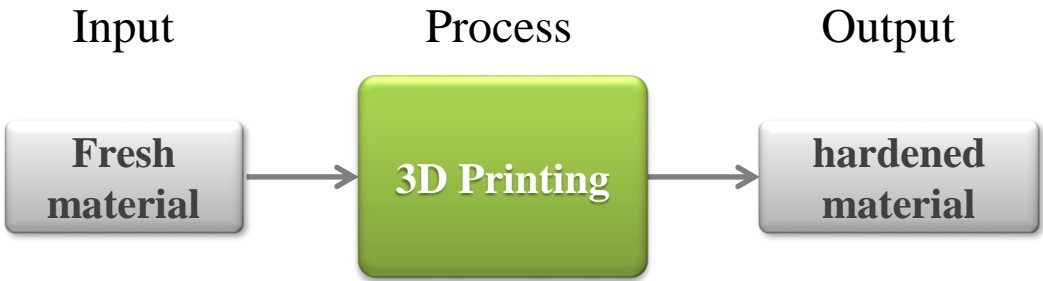

**Figure 1.** 3D printing process: input and output.

## 5. Proposed Theory-Based Approach for 3D Printing Process in Construction

The proposed approach comprises four steps:

- Step 1: Selection of sustainability indicators adapted with additive manufacturing technology (layer-by-layer deposition), such as porosity and sorption, and desorption isotherms.
- Step 2: Experimental campaign according to the procedures defined in this report, in the various standards, and from previous projects; e.g., the characterization of samples taken during the various printing tests.
- Step 3: Determination of the different sustainability indicators using existing correlations and empirical models. The objective is to study the impact of composition and implementation parameters on sustainability.
- Step 4: Prediction of the lifespan of the materials studied and presentation of recommendations for improving formulation and implementation.

This paper draws on the results of three printing campaigns that used printed cementitious matrix formulations. The first campaign used a conventional formulation. The second campaign used a slow-hardening formulation for the cementitious printed material, and the third campaign used a rapid-hardening formulation. Those formulations were compared thanks to a selection of durability

indicators. After that, a printing framework is proposed, and some recommendations for future studies are presented.

## 6. Research Methodology

Tests were conducted to evaluate the properties that influence the durability of cementitious matrix materials. These tests involved essential tools for comparing the performance of different formulations and for studying the impact of composition and implementation. The choice of these tests was performance-based.

The tests were performed on molded specimens or cores taken from samples of an unknown formulation. The durability tests were blinded, which limited the extent of analysis.

A great deal of interest was expressed in this report for the characterization of the porous structure, which has a key role in the durability of cementitious materials. As part of a performance-based approach, sustainability indicators were studied to assess and predict the durability of the materials. These parameters make it possible to determine the properties of materials in relation to the environment and to feed predictive models for aging.

The research methodology consisted of assessing the durability of cementitious printed materials. Samples were recovered after each printing campaign to perform durability tests and the results compared throughout the evolution of successive printing campaigns.

### 6.1. Materials

Three printing campaigns that used cementitious matrix formulations were realized. The first campaign used a conventional formulation (Figure 2). The second campaign used a slow-hardening formulation for the cementitious printed material, and the third campaign used a rapid-hardening formulation (Figure 3). All specimens were made under the same printing conditions (same pumping rate, without vibration and shock). The three campaigns are designated as follows:

- Campaign 1: MC14-10-16,
- Campaign 2: MCR19-01-17,
- Campaign 3: MCR20-01-17.

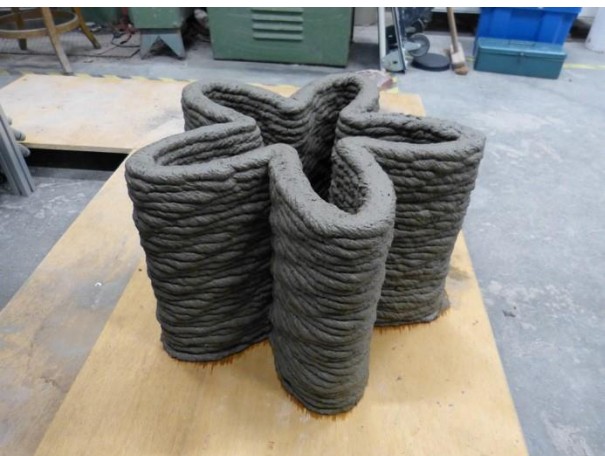

**Figure 2.** Image from the first 3D printing campaign that resulted in the MC14-10-16 samples. (French Project MATRICE [40]).

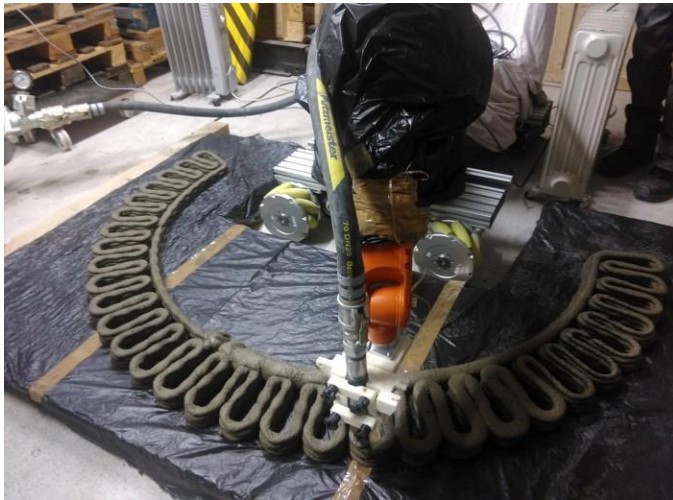
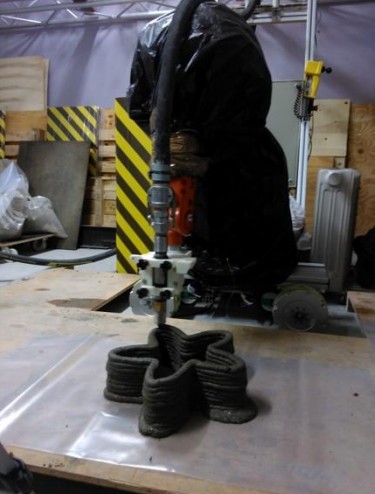

**Figure 3.** Images from the second 3D printing campaign that resulted in the MCR20-01-17 samples. (French Project MATRICE [40]).

## 6.2. Accessible Porosity to Water

Porosity is an indicator of concrete quality and is one of the most important indicators of sustainability. It has a direct impact on mechanical strength and durability. The porous network is responsible for the penetration and infiltration of aggressive substances into the concrete. Porosity is determined by hydrostatic weighing.

This test also determines water absorption and wet and dry densities (Figure 4).

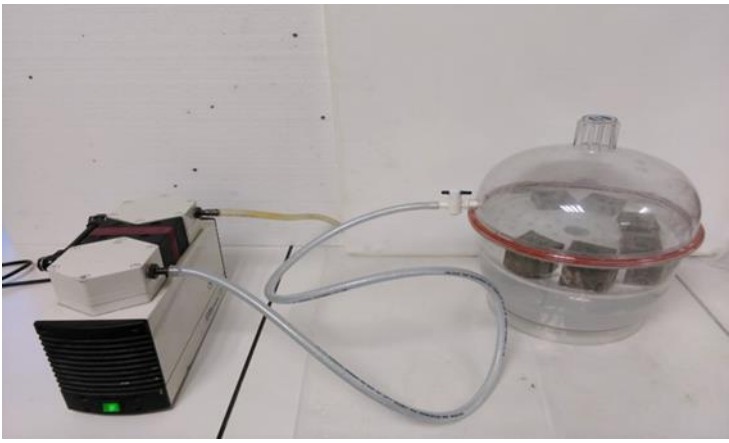

**Figure 4.** Vacuum specimen saturation.

## 6.3. Water Absorption

Water absorption is a measure of the amount of water absorbed through the concrete's pores open to the surrounding environment. It is determined by immersing a dry, weighed test piece in water and measuring the mass increase (see Figure 5). It is expressed as a percentage of the dry mass of the specimen. The water absorption is calculated using Equation (1):

$$Abs = \frac{m_{humide} - m_{seche}}{m_{seche}} * 100 \tag{1}$$

$M_{humide}$: constant wet mass of the specimen after immersion
$M_{sèche}$: constant dry mass of the specimen after drying in the oven

### 6.4. Wet and Dry Volumetric Masses

The wet (MVH) and dry (MVS) densities are calculated using the following expressions:

$$MVH = \frac{m_{humide}}{V} \tag{2}$$

$$MVS = \frac{m_{seche}}{V} \tag{3}$$

The volume ($V$) of the specimen is determined by hydrostatic weighing (Figure 5), thanks to the following formula:

$$V = \frac{m_{humide} - m_{seche-eau}}{\rho_w} * 100 \tag{4}$$

$m_{seche-eau}$: Underwater mass of the sample determined by hydrostatic weighing; $\rho_w$: density of water, 1000 kg/m$^3$.

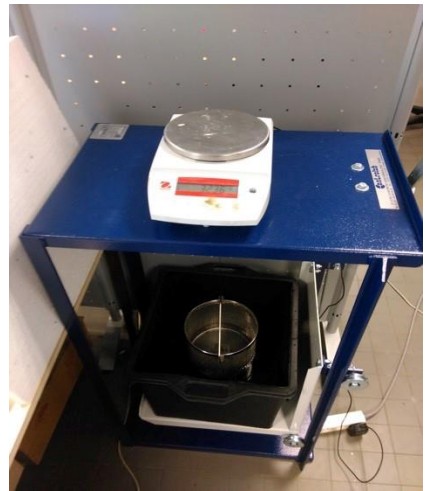

**Figure 5.** Hydrostatic weighing instrument.

### 6.5. Porosity

Porosity is determined by the following formula:

$$Abs = \frac{m_{humide} - m_{seche}}{m_{humide} - m_{seche-eau}} * 100 \tag{5}$$

### 6.6. Compressive Strength

Compressive strength is determined in accordance with NF EN 206 (2014) and measured on 28-day cubic samples (Figure 11). Tests were performed on five specimens for each formulation. Samples have a cubic form 8 × 8 cm.

### 6.7. Sorption–Desorption Isotherms

The purpose of this test is to determine the sorption–desorption isotherms of samples from the different printing campaigns for cementitious and earth-based materials. Apparel in Figure 6 was used.

The samples studied in this report are the cement paste samples recovered on October 14, 2016, as well as samples from the printing campaigns of January 19 and 20, 2017. The specimens were cut using a diamond disc saw to adapt them to this test.

The principle of the test is to determine the mass moisture content of the samples at different levels of relative humidity. In the beginning, the specimens were dried until a constant mass was obtained. Then, the specimens were placed in cups and left in a climatic chamber (Figure 13), programmable in

temperature and humidity. This allowed the specimens to be weighed automatically without being removed from the chamber by pre-defining the measurement conditions of, e.g., temperature and air velocity, and programming the desired relative humidity cycle. This reduced the time required to achieve equilibrium at the relative humidity (RH) RH under consideration because it does not disturb the environment and samples during weighing.

The temperature was kept constant (23 °C). RH levels of 30%, 50%, 75%, and 95% were tested. The maximum relative humidity was limited to 95% because the full range of humidity is difficult to achieve in practice.

The equilibrium mass obtained for each relative humidity considered was used to determine the mass water content of the sample in percent.

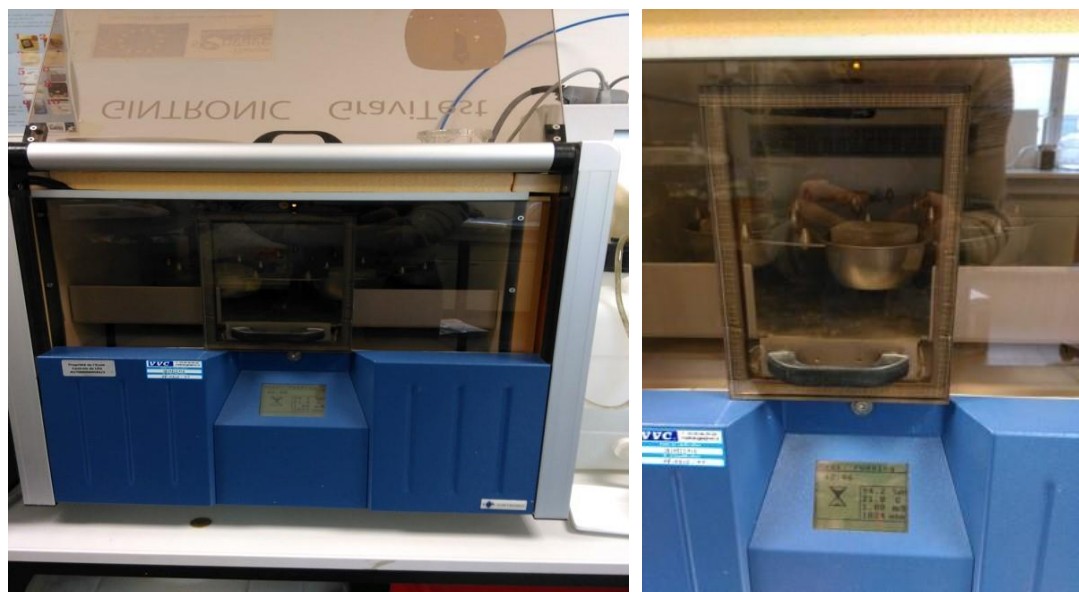

**Figure 6.** Gravitest apparatus for sorption–desorption isotherms.

## 7. Results and Discussion

Table 2 summarizes the results of the experiment. Water absorption, accessible porosity to water, and compressive strength of the three campaigns' printed samples are plotted as a radar graph that describes the durability performance of the printed materials.

**Table 2.** Water absorption, accessible porosity to water, and compressive strength of the three campaigns' printed samples.

| Durability indicators | MC14-10-16 | MCL19-01-17 | MCR20-01-17 |
|---|---|---|---|
| Compressive strength (%) | 39,67 | 29 | 11,35 |
| Water absorption (%) | 5,9 | 6,7 | 8,2 |
| Accessible Porosity to water (%) | 12,8 | 14,6 | 17,5 |

### 7.1. Water Absorption

The MC14-10-16 specimens from the first printing campaign of November 10, 2016 have good average potential durability because the average porosity value is 12.8%. However, for the MCL19-01-17 formulation, the average porosity is 14.6%. Thus, the potential durability is low for these specimens. However, for the rapid recovery formulation, durability is very low, with an average value of 17.5% (greater than 16%).

### 7.2. Compressive Strength

The compressive strength is higher for the specimens of the first printing campaign, with an average value equal to 39.67 MPa. However, for the rapid-setting formulation of January 20, 2017, a significant deterioration in compressive strength was observed that reached an average value of 11.35 MPa.

### 7.3. Sorption and Desorption Curves

Experimental measurements are used to plot the sorption and desorption curves, or isotherms of the specimens studied as shown in Figure 7. Knowledge of the moisture content for each relative humidity provides a point on the curve.

For the desorption curve, the samples are placed successively in a series of test environments where the relative humidity decreases in stages. The starting point of this curve corresponds to RH = 95%. Figure 7 shows the sorption/desorption curves obtained at 23 °C. The mass moisture content is defined as the ratio of the evaporable water mass to the dry material mass.

The results show that MC14-10-16 molded specimens have a low hygroscopic power that does not exceed 1%. However, the specimens recovered from the test bodies printed during the printing campaigns of January 19 and 20, 2017 have a higher hygroscopic power, especially for fast-setting samples with a mass moisture content of 1.6% at RH = 95%. This makes these test bodies more accessible to aggressive agents than molded samples.

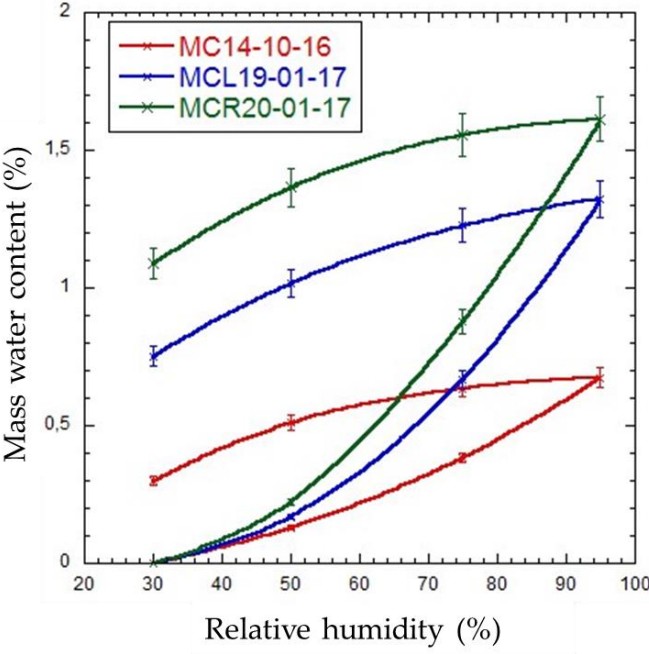

**Figure 7.** Sorption/Desorption isotherms of cementitious material specimens.

The isotherms of sorption and desorption show the existence of the phenomenon of hysteresis, which reflects the fact that it is easier for water to enter the porous network than to leave it. This is frequently explained by the geometric shape of the pores, with voids being connected by smaller pass sizes. Indeed, the results show that for the MCR20-01-17 tests during the desorption phase, samples keep a mass moisture content equal to 1.1% at RH = 30%. This will lead to deterioration in the durability of this formulation compared with others.

The results will provide insights into to the specific surface area according to BET theory [41], as well as information on pore size distribution by applying the BJH theory [42].

### 7.4. Analysis and the Proposed Durability Approach for 3D-Printed Materials

While the three printing campaigns resulted in printed structures, the results reveal different characteristics. This led us to rethink the indicators for 3D-printed materials for construction due to the performance-based approach. Those indicators are associated with fresh, pre-hardened, and hardened material as shown in Figure 8.

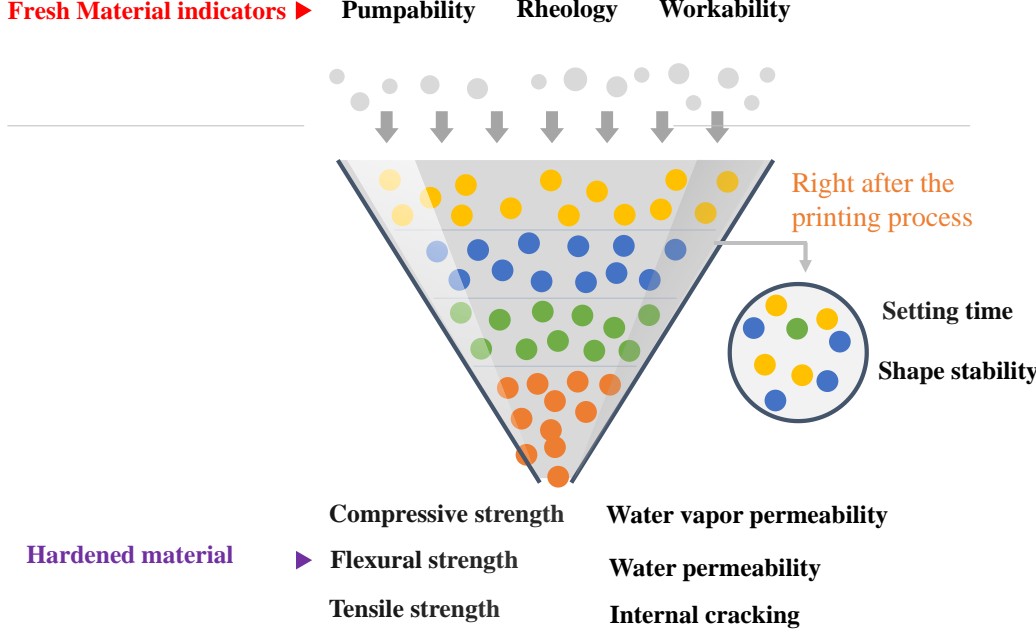

**Figure 8.** General durability framework for 3D-printed construction materials.

### 7.5. Central Role of Rheology as an Indicator

A material's constituents have a direct influence on its rheology. Having an idea of the rheological properties of concrete allows for good control of the pre-mixing flow rate, which is decisive during printing. In other words, rheology defines the flow velocity for given shear stress.

Different categories of fluids exist with the most known being the Newtonian fluids. Regardless of the stress applied, these fluids do not have a shear threshold. Their flow is proportional to the stress and keeps a constant viscosity as shown in Figure 9.

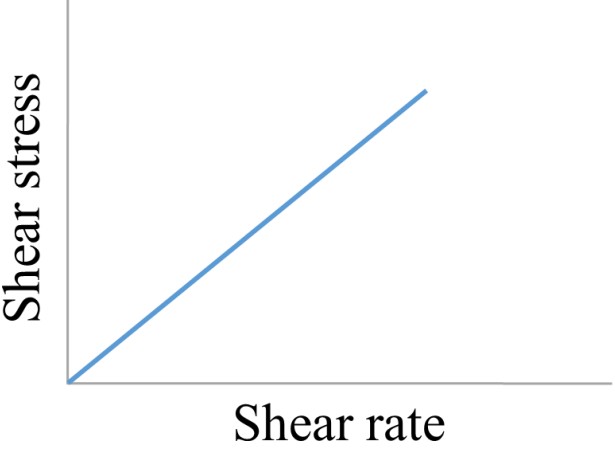

**Figure 9.** Behavior of a typical Newtonian fluid.

Construction materials, such as concrete, are generally non-Newtonian fluids before hardening. Their curve is set, and literature gives a holistic view on how construction materials behave. The challenge for the scientific community regarding 3D-printed materials is to understand how printed materials behave before and after the printing process. Figure 10 presents different types of non-Newtonian fluids. Those fluids are characterized not by traditional indicators, such as compressive strength, but by indicators, such as viscosity.

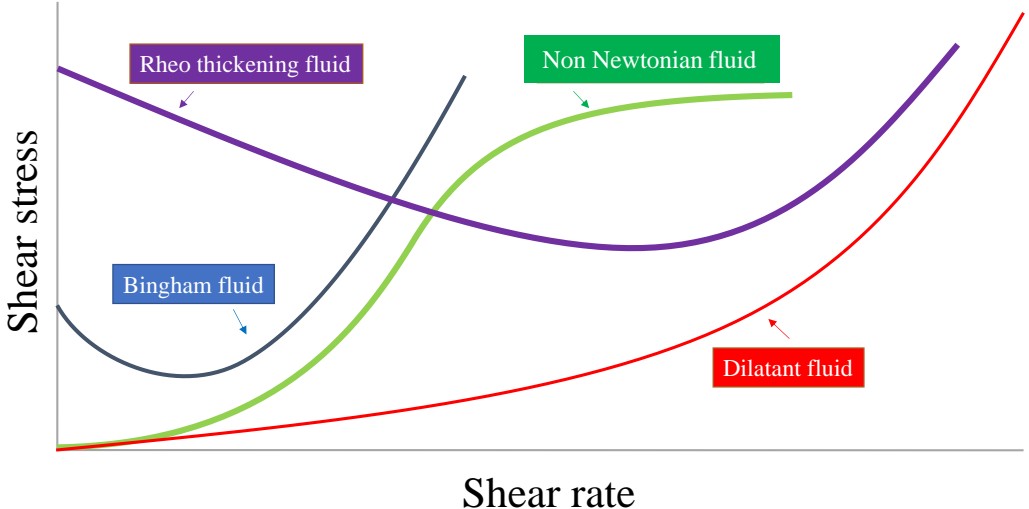

**Figure 10.** Shear behavior for different types of fluids.

In its fresh state, concrete is considered a non-Newtonian rheofluidifying fluid, i.e., its viscosity will decrease, and it will harden slowly if not subjected to stress.

Printed concrete is not like poured concrete in that it tends to have a low E/C ratio and the grains have a small diameter and spindle grain size. In his article on the properties of printable cementitious materials, Suvash Chandra Paul used a rheometer called the "Schleibinger Viscomat NT" to measure the rheology of printable mortar formulations [37]. However, the critical point of this test is that it cannot directly determine viscosity and shear stress. Following this, Wangler et al. were able to determine the calibration coefficients for the values obtained from the Viscomat and transformed them into shear stress and plastic viscosity [9].

### 7.5.1. Pumpability Indicator

Concrete is said to be pumpable if, under pressure, the flow is enough to make the printing process smooth. The E/C ratio plays a role in determining the pumpability. A pumpable material is not necessarily printable. The material can be pumpable but not adequate for printing in terms of durability and shape.

### 7.5.2. Shape Stability

Shape stability is a synonym of what researchers call "buildability." Shape stability quantifies the number of filament layers that could be constructed without significant deformation of the lower layers. It must be possible to indicate whether the layered structure is able to predict failure time when the structure collapses. Indeed, to learn more about this parameter, the shape stability and resistance of the layers come into play. For shape stability, contour crafting has been adopted (e.g., cylinder stability test), which saves us printing tests with the layer settlement test.

### 7.6. Anisotropy of 3D-Printed Construction Materials

Anisotropy is the property of materials that have different characteristics depending on their orientation. Wood, for instance, is anisotropic because of its compressive strength changes according to the orientation of the constraint (i.e., wood grain).

The printing process of construction materials introduces anisotropy. The materials are deposited layer-by-layer, which creates potential weaknesses between the layers that should be studied in depth in future research. Figure 11 shows how vertical and horizontal constraints can affect the printed layers.

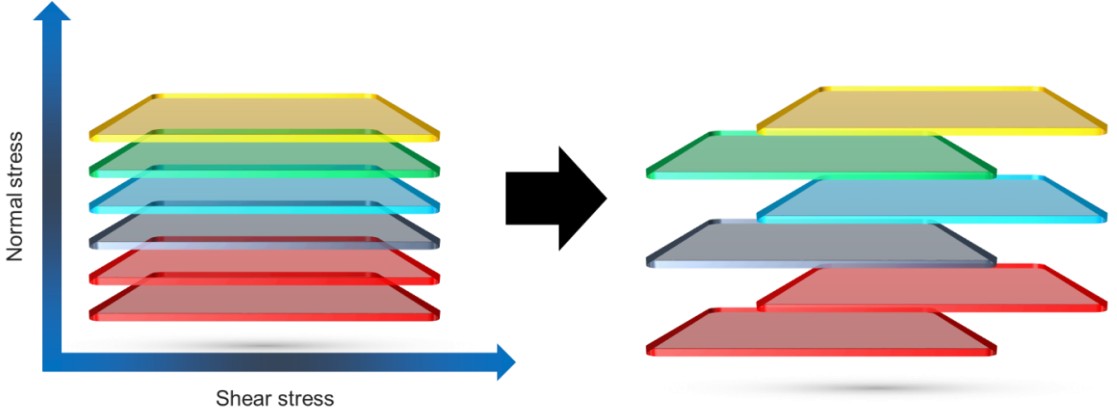

**Figure 11.** Vertical and horizontal constraints applied to printed layers.

The anisotropic behavior of printed construction materials is introduced during the layer-by-layer deposit process as opposed to other setting methods, such as casting [43]. Thus, our ascertainment is that durability indicators should be acknowledged for both horizontal and vertical variations.

### 7.7. Methods to Mitigate the Shear Stress Generated by the Printing Process

The first method is the use of fibers to link the layers as shown in Figure 12. Thus, the shear stress could be significantly reduced, and the structural state of the 3D-printed structure can stand a chance to offset the constraints.

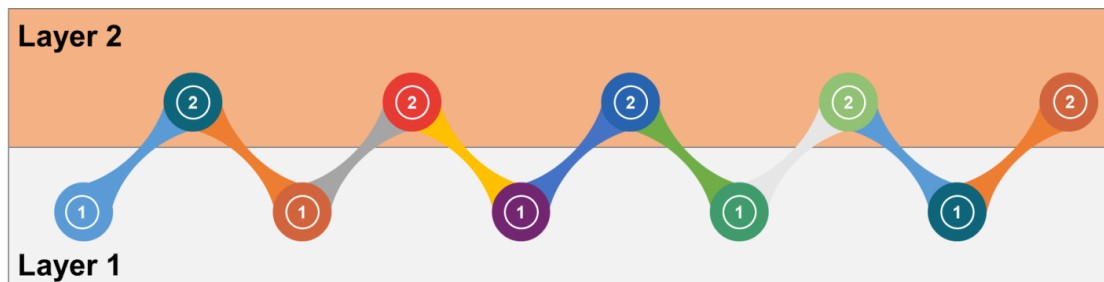

**Figure 12.** Fiber linkage of layers.

The second method is the use of a shear consolidation tube as a linkage (Figure 13). This method is to be further developed in future research.

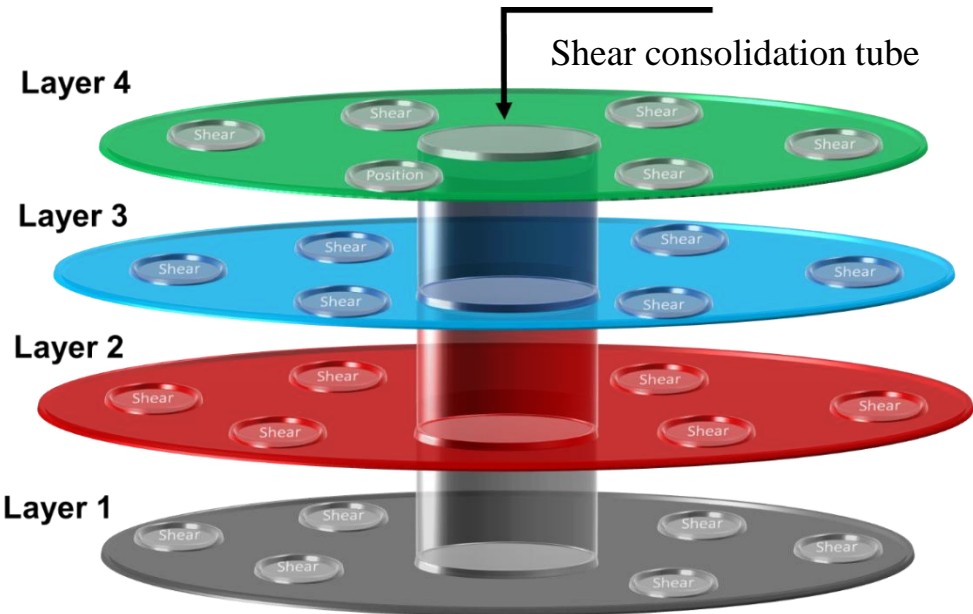

**Figure 13.** Shear consolidation tube as a linkage mechanism.

## 8. Conclusions

The objective of this research was to evaluate the properties that influence the durability of printed construction materials. These tests involved durability indicators for comparing the performance of three tested formulations. The choice of these indicators was based on a performance-based approach rather than a prescriptive one. Three printing campaigns were realized, and the results compared the performance of the samples. Indicators, such as porosity and water absorption, were identified for the performance-based approach. The results led to the proposition of some performance indicators to consider when evaluating printed construction materials. Rheology, pumpability, and workability are the indicators identified for the input printing process. For the output (hardened materials), the indicators are compressive strength, water vapor permeability, flexural strength, water permeability, tensile strength, and internal cracking. Those indicators should be assessed on both the vertical and horizontal axis because of the anisotropy of the printed materials. Future research should focus on testing the framework, and the durability indicators that help assess the pre-printability of construction materials. Indeed, researchers need a macro-way to evaluate the printability. Those tests are a pre-assessment and not the final assessment of printability.

**Author Contributions:** Conceptualization, Z.L.; Methodology, Z.L..; Analysis: Z.L. & Z.D.; writing-original draft preparation, Z.D.; writing-review and editing, Z.L. & Z.D.; funding acquisition, Z.L.

**Funding:** This research work has been carried out in the frame of the MATRICE Project, co-funded by the region "Hauts de France" and the European Union with the European Regional Development Fund.

**Conflicts of Interest:** The authors declare no conflict of interest.

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
