# Peer review of "Performance Indicators of Printed Construction Materials: a Durability-Based Approach"

_buildings, doi:10.3390/buildings9040097_

Round 1
Reviewer 1 Report
This paper has lot of flaws which needs to correct before it can be considered for publication. The following comments should consider to improve the paper.
1. In introduction, please discuss briefly about the 3D printing methods, materials, limitation, future, etc. See this paper where some of these are discussed broadly: A review of 3D concrete printing systems and materials properties: current status and future research prospects. Rapid Prototyping Journal, 2018, 24(4), 784-798.
2. What are the new things this paper has discovered? There are so many existing papers already dealt with similar kind of work. So, please highlight the major outcomes of your research.
3. Give some information about the compressive strength test such as size of samples, testing directions on printed samples, etc.
4. What was the pump speed, materials flow rate?
5. 4. Page 7, line 213, what does it mean by demoudling? 3D printing doesn’t need any formwork.
6. Figure 7 is confusing. You may consider to use table for the presentation of results.
7. Page 10, line 291, where is Figure 14 in the paper? Also I noticed there are many figures which are not referenced in the text.
8. Fig 8, could you please explain why the each lines are coming downward?
9. After figure 8 in page 11, page 12 starts with figure 1 again!!
10. Fig 2 not necessary to show here. Where is the reference for this figure?
11. Reference for Fig 3?
12. Figure 5, how fiber can link this way? Please explain a bit. Similarly, figure 6 also needs some explanation.
13. There are limited relevant references have been cited in this paper. Please search for the 3D concrete printing related papers, read and cite here.
https://doi.org/10.1016/j.measurement.2017.08.051
https://doi.org/10.1016/j.acme.2017.02.008
https://doi.org/10.1016/j.cemconcomp.2018.10.002
https://doi.org/10.1016/j.compositesb.2018.11.109
14. Conclusion needs to rewritten. At current stage it is not clear what the major outcomes from this research are.
Author Response
1. In introduction, please discuss briefly about the 3D printing methods, materials, limitation, future, etc. See this paper where some of these are discussed broadly: A review of 3D concrete printing systems and materials properties: current status and future research prospects. Rapid Prototyping Journal, 2018, 24(4), 784-798.
We ‘ve added some literature review about 3D printing.
Here is an excerpt:
It is in this context that the durability of materials derived from additive manufacturing technology will be studied in this research. 3D printing in construction is a new field of study that attracted many researchers [1]–[3]. The current challenges of 3D printing concerns the robotics systems, the software part [4], and the material part[5], [6]. The key challenges for printability were outlined by Wangler et al. [4]. The focus was the concrete extrusion and intermix with the previously deposited layer. Each layer must support its own weight and the weight of the material to be subsequently deposited.
Example of research projects on tensile strength of 3D printed materials for construction [18]–[20]. The concept of printability was also investigated in literature research, and linked to some indicators such as the shear rate, viscosity, and thixotropy [21].
2. What are the new things this paper has discovered? There are so many existing papers already dealt with similar kind of work. So, please highlight the major outcomes of your research.
This paper draws on the past results of researchers in 3D Printing for construction, by providing results of tests on 3 printing campaigns. Those campaigns used different printing formulations and compared them. So, the main contributions is the comparison of 3 different formulations using durability indicators.
This main contribution helped suggest a framework that could be used in the future to assess a printed formulation
3. Give some information about the compressive strength test such as size of samples, testing directions on printed samples, etc.
This sentence was added to the manuscript. “Samples have a cubic form 8x8 cm. »
4. What was the pump speed, mater ials flow rate?
The pump speed (which is the flowrate) for the research undertaken varied from: 0,1 to 4 l / minutes.
The reason for this variation is the experimental setup that is not to point for all 3D printing experimentations. The objective of this paper is to set up pre-indicators so that the material flows with a fixed rate (which is not achieved till now)
5. 4. Page 7, line 213, what does it mean by demoudling? 3D printing doesn’t need any formwork.
Yes. We removed the sentence “After demolding, the samples were stored in water at 20°C in the laboratory” from this section.
The demoulding concerns the samples that were used for the compressive tests (after the coring of the printed product).
6. Figure 7 is confusing. You may consider using table for the presentation of results.
The authors have changed the figure to Table 1 as follow:
Table 1. Water absorption, accessible porosity to water, and compressive strength of the three campaigns’ printed samples
Durability indicators | MC14-10-16 | MCL19-01-17 | MCR20-01-17 |
Compressive strength (%) | 39,67 | 29 | 11,35 |
Water absorption (%) | 5,9 | 6,7 | 8,2 |
Accessible Porosity to water (%) | 12,8 | 14,6 | 17,5 |
7. Page 10, line 291, where is Figure 14 in the paper? Also I noticed there are many figures which are not referenced in the text.
The referencing of the figures is now well done. Figure 14 is now put in the paper with a proper referencing.
8. Fig 8, could you please explain why each lines are coming downward?
Figure 8 shows the sorption – desorption isotherms. In literature, the sorption curve in upward. This is due to the increase in the level of mass water content (and thus the increase in humidity). The desorption isotherm is downward and shows how the material behaves when dropping humidity levels.
9. After figure 8 in page 11, page 12 starts with figure 1 again!!
Yes. Sorry for that. We corrected the figure’s sequence. Now the article is to the point.
10. Fig 2 not necessary to show here. Where is the reference for this figure?
Following the right sequencing of figures, figure 2 is now called Figure 9. The reference of figure 9 is in the paragraph above the figure, as follow:. Their flow is proportional to the stress and keeps a constant viscosity as shown in Figure 9
11. Reference for Fig 3?
We made figure 3 based on the literature that defines the types of existing fluids. This literature is explored in the article before showing the figure.
For information, Figure 3 is not called 3 anymore thanks to the right figure sequencing.
12. Figure 5, how fiber can link this way? Please explain a bit. Similarly, figure 6 also needs some explanation.
This part builds upon what we found precedingly about the durability indicators. We found that the compressive strength of 3D printed materials is relatively good. The next problem is shear rate.
Many research articles address the shear rate as a critical issue for 3D printing, such as this article.
Such as this article: https://doi.org/10.1016/j.compositesb.2018.11.109
The issue with 3D printed materials is that the resistance between the layers. In another word, the layers don’t hold together, or at least, they are easily disconnected.
For that, we proposed method to fix this issue. Fibers can be added to link the layer.
Also, a tube that go vertically could be placed before the printing process. Its role is to be a pillar that links the layers.
13. There are limited relevant references have been cited in this paper. Please search for the 3D concrete printing related papers, read and cite here.
https://doi.org/10.1016/j.measurement.2017.08.051
https://doi.org/10.1016/j.acme.2017.02.008
https://doi.org/10.1016/j.cemconcomp.2018.10.002
https://doi.org/10.1016/j.compositesb.2018.11.109
Thank you for providing all those really interested articles. They are totally in line with our contribution and vision. We integrated them to the manuscript.
14. Conclusion needs to rewritten. At current stage it is not clear what the major outcomes from this research are.
The conclusion was modified according to the reviewer’s comments. Here is the modified conclusion:
The objective of this research was to evaluate the properties that influence the durability of printed construction materials. These tests involved durability indicators for comparing the performance of three tested formulations. The choice of these indicators was based on a performance-based approach rather than a prescriptive one. Three printing campaigns were realized and the results compared the performance of the samples. Indicators such as porosity and water absorption were identified for the performance-based approach. The results let to the proposition of some performance indicators to consider when evaluating printed construction materials. Rheology, pumpability, and workability are the indicators identified for the input printing process. For the output (hardened materials), the indicators are compressive strength, water vapor permeability, flexural strength, water permeability, tensile strength, and internal cracking. Those indicators should be assessed on both the vertical and horizontal axis because of the anisotropy of the printed materials. Future research should focus on testing the framework, and the durability indicators that helps assess the pre-printability of construction materials. Indeed, researchers needs a macro-way to evaluate the printability. Those tests are a pre-assessment, and not the final assessment of printability.

Reviewer 2 Report
The objective of this study was to examine the printed construction materials.
The study is interesting but it needs some changes before publication:
A common practice in scientific papers is to include a brief paragraph at the end of the Introduction in order to indicate the structure of the document. This helps the reader to have an accurate idea about the organization and facilitates the reading.
As far as the formal aspect of the article is concerned, I think there is one fundamental point missing from the article: the conclusions. The article, although it includes some final notes, does not have a specific point for conclusions.
Another important aspect to take into account is the bibliography, there is a lack of references from authors who have reinforced the premises of the article.
A discussion comparing with other authors is needed.
In the end of the manuscript I think is important to present the recommendations for future studies.
References should be expanded and a larger number of the MDPI group should be entered.
Author Response
A common practice in scientific papers is to include a brief paragraph at the end of the Introduction in order to indicate the structure of the document. This helps the reader to have an accurate idea about the organization and facilitates the reading.
The following paragraph is added right before the methodology section:
This paper draws on the results of three printing campaigns that used printed cementitious matrix formulations. The first campaign used a conventional formulation. The second campaign used a slow-hardening formulation for the cementitious printed material and the third campaign used a rapid-hardening formulation. Those formulations were compared thanks to a selection of durability indicators. After that, a printing framework is proposed and some recommendations for future studies are presented.
As far as the formal aspect of the article is concerned, I think there is one fundamental point missing from the article: the conclusions. The article, although it includes some final notes, does not have a specific point for conclusions.
The reason the paper is missing punctual conclusions is that the final aim is to propose a useful framework to have a printable material. Here is a text that was added to see the point:
It is in this context that the durability of materials derived from additive manufacturing technology will be studied in this research. 3D printing in construction is a new field of study that attracted many researchers [1]–[3]. The current challenges of 3D printing concerns the robotics systems, the software part [4], and the material part [1], [5]. The key challenges for printability were outlined by Wangler et al. [4]. The focus was the concrete extrusion and intermix with the previously deposited layer. Each layer must support its own weight and the weight of the material to be subsequently deposited.
For that the article draws on the results of 3 printing campaigns. Each campaign is linked to a different printing formulation.
Durability indicators were set, and the formulations compared according to those indicators.
Those steps helped build the framework.
Another important aspect to take into account is the bibliography, there is a lack of references from authors who have reinforced the premises of the article.
We’ve added a series of references to the articles of previous research projects on 3D printing in construction.
A discussion comparing with other authors is needed.
Other authors were considered for this research study. More references were added to point out the previous contribution of authors.
Also paragraphs were added for that. Here is an example:
Example of research projects on tensile strength of 3D printed materials for construction [18]–[20]. The concept of printability was also investigated in literature research, and linked to some indicators such as the shear rate, viscosity, and thixotropy [21].
In the end of the manuscript I think is important to present the recommendations for future studies.
Yes. The following text was added to the paper.
Future research should focus on testing the framework, and the durability indicators that helps assess the pre-printability of construction materials. Indeed, researchers needs a macro-way to evaluate the printability. Those tests are a pre-assessment, and not the final assessment of printability.
References should be expanded and a larger number of the MDPI group should be entered.
References were added. Examples of MDPI references added:Here are some MDPI references that were added:
H. Jeong et al., “Rheological Property Criteria for Buildable 3D Printing Concrete,” Materials (Basel)., vol. 12, no. 4, p. 657, Feb. 2019.
S. Bong et al., “Method of Optimisation for Ambient Temperature Cured Sustainable Geopolymers for 3D Printing Construction Applications,” Materials (Basel)., vol. 12, no. 6, p. 902, Mar. 2019.

Reviewer 3 Report
The article would like to deal with the performance indicators of 3D-printed materials from a durability point of view. The paper is very confused, the topic is not introduced exhaustively and the experimental part is very poor. Moreover, results are poorly presented and the discussion is a bit confused.
For these reasons, I am forced to reject the paper.
Author Response
The article would like to deal with the performance indicators of 3D-printed materials from a durability point of view. The paper is very confused, the topic is not introduced exhaustively and the experimental part is very poor. Moreover, results are poorly presented and the discussion is a bit confused.
For these reasons, I am forced to reject the paper.
Thanks to the reviewer’s comments, the paper was improved drastically.
The reason the paper is confused is that the final aim is to propose a useful framework to have a printable material. Here is a texte that was added to see the point:
It is in this context that the durability of materials derived from additive manufacturing technology will be studied in this research. 3D printing in construction is a new field of study that attracted many researchers [1]–[3]. The current challenges of 3D printing concerns the robotics systems, the software part [4], and the material part [1], [5]. The key challenges for printability were outlined by Wangler et al. [4]. The focus was the concrete extrusion and intermix with the previously deposited layer. Each layer must support its own weight and the weight of the material to be subsequently deposited.
For that the article draws on the results of 3 printing campaigns. Each campaign is linked to a different printing formulation.
Durability indicators were set, and the formulations compared according to those indicators.
Those steps helped propose the framework.

Round 2
Reviewer 1 Report
Please improve your introduction by considering 3D printing material, methods and future research direction. Also section 3 needs to improve a bit.
Author Response
We ‘ve added some literature review about 3D printing.
Here is an excerpt:
It is in this context that the durability of materials derived from additive manufacturing technology will be studied in this research. 3D printing in construction is a new field of study that attracted many researchers [1]–[3]. The current challenges of 3D printing concerns the robotics systems, the software part [4], and the material part[5], [6]. The key challenges for printability were outlined by Wangler et al. [4]. The focus was the concrete extrusion and intermix with the previously deposited layer. Each layer must support its own weight and the weight of the material to be subsequently deposited.
Example of research projects on tensile strength of 3D printed materials for construction [18]–[20]. The concept of printability was also investigated in literature research, and linked to some indicators such as the shear rate, viscosity, and thixotropy [21]

Reviewer 2 Report
A good job has been done to adapt the article to the demands. I think it is necessary to add more bibliographic references in points 1 and 2 to finally be accepted.
Author Response
Number of references changed from 26 to 41.
Examples:
The concept of printability was also investigated in literature research, and linked to some indicators such as the shear rate, viscosity, and thixotropy [25], [26].
Another important parameter in determining sustainability indicators is the water status of the samples, which is essential for the development of chemical reactions and their macroscopic consequences [28]–[30].

Reviewer 3 Report
The paper was greatly improved as a consequence of revision. I have only a comment about the introduction (see the attached file). For these reasons, the paper can be accepted for publication after minor revision.

Author Response

(The authors gave the same response as above.)
